# Small target detection algorithm based on the fusion attention mechanism and multi-layer convolution

Xiujing Li[ID]©*, Haifei Zhang©, Yiliu Hang©, Hao Chen©

Information Engineering, Nantong Institute of Technology, Nantong, Chongchuan, China

© These authors contributed equally to this work.
* 982888184@qq.com

## Abstract

In the realm of unmanned aerial vehicles, we proposed an enhanced small target detection algorithm, MGAC-YOLO, to address the challenges of missed detections and low accuracy associated with small target identification. Initially, we designed the MConv (Multi-layer Convolution) module to replace the conventional Conv module within the backbone network, thereby augmenting the dimensionality of information capture and enhancing the detection performance for small targets. Subsequently, we harnessed the advantages of both attention mechanisms—GAM (Global Attention Mechanism) and CloAttention (Contextualized Local and Global Attention)—to create a GACAttention module that extracts small target features from both global and local perspectives, thereby enriching the network's focus on small target feature information and further enhancing its feature processing capabilities. Finally, we incorporated an additional small target detection layer to capture feature information at a shallower level, thereby reducing the likelihood of missed detections and bolstering the detection capabilities for small targets. Experimental results on the VisDrone2019 dataset demonstrate that the Precision, $mAP_{50}$, and $mAP_{50-95}$ of the MGAC-YOLO algorithm have improved by 5.3%, 6.3%, and 4.4%, respectively, in comparison to the baseline model YOLOv8s. Furthermore, when compared to other leading algorithms, the MGAC-YOLO algorithm has exhibited notable superiority.

## Overview

Object detection stands as a fundamental task within the domain of computer vision, gaining significant prominence alongside the rapid advancements in artificial intelligence technology. This technology enables the identification of specific objects within images or videos, as well as the determination of their locations and dimensions. Its applications span various sectors, including security surveillance, autonomous vehicle navigation, and medical imaging analysis. With the ongoing evolution of unmanned aerial vehicle (UAV) technology, object detection from UAV perspectives

**Data availability statement:** The data underlying the results presented in the study are available from Baidu Netdisk (Link: https://pan.baidu.com/s/1kAgxuxByfz8AaY0RFLl9gQ?pwd=l7q9).

**Funding:** The author(s) received no specific funding for this work.

**Competing interests:** The authors have declared that no competing interests exist.

has revealed distinct application potential, attributed to its unique aerial vantage point and agile mobility. UAVs can efficiently survey extensive areas, facilitating operations such as traffic surveillance, environmental assessment, and disaster evaluation.

The evolution of object detection algorithms has transitioned from traditional feature-based approaches to those powered by deep learning. Deep learning-based object detection algorithms are primarily categorized into two main types: two-stage detection algorithms and one-stage detection algorithms. Two-stage detection algorithms, which include R-CNN [1], Faster R-CNN [2], and others, execute the object detection process in two phases: generating candidate regions followed by region classification and localization. While these algorithms achieve relatively high detection accuracy, they tend to exhibit lower efficiency, making them less suitable for real-time monitoring applications. In contrast, one-stage detection algorithms, such as SSD [3] (Single Shot MultiBox Detector), the YOLO series [4] (You Only Look Once), and RetinaNet [5], directly extract features from images to accomplish object detection tasks. This approach offers superior detection speed, rendering it more appropriate for real-time application scenarios.

With the advancement of deep learning technologies, significant progress has been made in the realm of object detection, particularly in areas such as multi-object detection, transfer learning, and salient object detection. Within the domain of object detection utilizing unmanned aerial vehicles (UAVs), several challenges persist. These include the limited pixel representation of targets in aerial imagery, the dynamic complexity of image scenes, variations in shooting scales and perspectives, and the presence of dense targets and occlusions [6]. Current research efforts primarily concentrate on multi-scale feature fusion and the refinement of attention mechanisms to mitigate these issues. In the study by LIU Qingqing et al. [7], an enhanced SSD algorithm was presented, incorporating multi-scale feature fusion and a spatial pyramid pooling module to bolster the model's efficacy in small-scale target recognition. BiEO-YOLOv8s [8] employed a bidirectional weighted multi-scale neck network (BiEO-Neck) to integrate shallow and deep features. BDH-YOLO [9] optimized multi-scale features by integrating BiFPN with a dynamic detection head (DyHead). JI Qin et al. [10] designed a PMSE module to enhance shallow features using parallel multi-scale dilation convolution. LIU H et al. [11] developed an efficient spatiotemporal interaction module (ESI), incorporating spatial pyramid convolution and recursive gated convolution to improve higher-order spatial semantic interaction. UAV-DETR [12] addressed feature map misalignment via dynamic grid sampling (SAC module). MPE-YOLO [13] enhanced stability in complex backgrounds through adversarial training and dynamic noise suppression mechanisms. YOLOv8-DEL [14] introduced a lightweight shared convolutional detection head, reducing parameters while boosting detection speed. SSC-YOLOv8n [15] proposed a coordinate attention dynamic decoupled head method, significantly improving the algorithm's sensitivity to positional information. Despite these advancements, current methodologies continue to exhibit limitations, including inadequate cross-scale interaction (resulting in deficient extraction of small target details), feature confusion in dense occlusion scenarios, and information loss during downsampling operations.

To further explore the application potential of UAVs in object detection, this paper proposes the MGAC-YOLO algorithm based on the YOLOv8s network as the baseline, aiming to improve the detection accuracy for small targets in UAV scenarios. The term "small targets" in this context refers to objects that occupy a small number of pixels in the image, or those exhibiting diminutive dimensions relative to the overall image and exhibit low-resolution characteristics. The algorithmic improvements are outlined as follows:

(1) To address the challenges of insufficient feature representation in shallow layers and poor training stability in deep networks for small target detection, this study designs an MConv multi-layer convolutional structure based on residual learning. This module significantly expands the receptive field by stacking depthwise separable convolutional layers, enabling the network to capture multi-scale spatial features. By introducing cross-layer residual connections [16], gradient propagation shortcuts are constructed, effectively alleviating gradient vanishing/explosion phenomena in deep networks.

(2) To overcome the limitations of traditional attention mechanisms, such as insufficient global context modeling and coarse-grained attention to local details, this paper proposes the GACAttention cross-dimensional attention module. This module innovatively integrates the global channel attention of GAM [17] and the local spatial attention of CloAttention [18]. The input feature map is processed through the channel and spatial attention branch modules respectively, thereby obtaining more comprehensive and rich small target features and improving the model's detection ability for small targets..

(3) To address the severe loss of small target semantic information in deep feature maps of the YOLOv8 network, this study proposes a multi-scale feature fusion detection strategy. By adding a small target detection head in shallow network layers, the strategy directly preserves high-resolution detail features and fuses them with deep-layer features through cross-level fusion. This approach leverages the pixel-level localization advantages of shallow features and the semantic discriminability of deep features, constructing a complementary feature representation to enhance detection robustness for small targets.

## Introduction to the YOLOv8 algorithm

YOLOv8 [19] represents a cutting-edge real-time object detection algorithm. It builds upon the foundation of YOLOv5, offering enhancements across various dimensions. Through its innovative network architecture and loss function design, YOLOv8 successfully improves detection accuracy while preserving high processing speed. Its performance on numerous standard datasets exceeds that of most existing object detection algorithms, underscoring its practicality and effectiveness.

The Backbone, which serves as the core network, forms the foundation of the entire object detection framework. It comprises multiple convolutional and pooling layers, utilizing residual connections and bottleneck structures. This architecture effectively extracts rich feature information from input images while minimizing network size and enhancing performance. Structurally, this component employs the C2f [20] module, which features fewer parameters and superior extraction capabilities, replacing the C3 module found in YOLOv5. Additionally, depthwise separable convolution and dilated convolution are implemented to further bolster feature extraction capabilities. A structural comparison between the C3 module and C2f module is illustrated in Fig 1.

The Neck, situated between the Backbone and the Head, functions as a critical intermediary. It not only transmits features extracted by the backbone network but also enriches the diversity and richness of feature information through its SPPF [21] module (Spatial Pyramid Pooling Fast), PAA [22] module (Probabilistic Anchor Assignment), and PAN [23] module (Path Aggregation Network). This enhances the model's detection capabilities for targets of varying scales and improves the overall training efficacy.

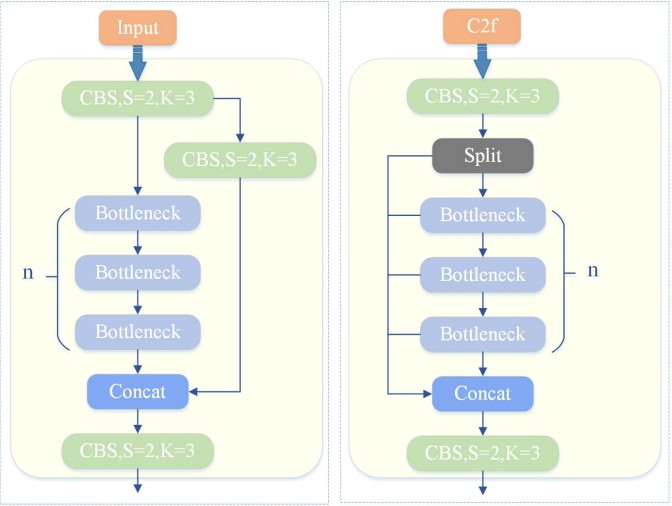

**Fig 1. Structural comparison diagram of the C3 module and the C2f module.**

The Head, located at the forefront of the network, plays a pivotal role in object detection. It is tasked with further processing the feature information relayed from the neck network and executing the final object detection and classification tasks. The Head component of YOLOv8 encompasses both a detection head and a classification head, which are responsible for predicting the bounding box regression values of anchor boxes, assessing the confidence of object presence, and managing multi-class tasks. Furthermore, the Anchor-free mechanism is employed to streamline the hyperparameter settings for anchor boxes, while the enhanced adaptive NMS [24] algorithm is utilized to mitigate false positives and missed detections, thereby improving detection accuracy.

The loss function of YOLOv8 is crucial for the algorithm's performance, as it encompasses not only the classification errors of objects but also the regression errors associated with bounding boxes. The YOLOv8 loss function is composed of several components, including classification loss (VFL Loss) [25], regression loss (CIOU [26] + DFL), localization loss, and object confidence loss, among others. Notably, VFL employs an asymmetric weighting mechanism that emphasizes positive samples over negative ones. DFL facilitates the network's rapid adaptation to the positional distribution near the target by modifying the positional modeling of the bounding boxes.

In this study, YOLOv8s serves as the foundational model due to its reduced parameter count and its effective balance between speed and accuracy, upon which enhancements are implemented.

### The MGAC-YOLO algorithm

The detection datasets derived from UAV aerial photography present challenges such as small object features, significant scale variations, dense background element distributions, and fluctuating viewing angles and lighting conditions. These factors contribute to issues like missed detections and diminished detection accuracy. To address the challenges of missed detections and low accuracy in small target detection by UAVs, this paper proposes enhancements based on the YOLOv8s architecture. Initially, the newly designed MConv module replaces the Conv module in the backbone network, employing multiple convolutional layers to enhance information capture dimensions while utilizing residual connections to mitigate vanishing and exploding gradient issues. Additionally, a novel GACAttention module is integrated before the SPPF module to extract small target features from both global and local perspectives, thereby enriching the network's focus on small target feature information. Finally, a dedicated small target detection layer is incorporated to capture

feature information from the shallower layers of the network, thereby bolstering the detection capabilities for small targets. The structure of the MGAC-YOLO algorithm is illustrated in Fig 2.

**The MConv convolution module**

The conventional convolution module, Conv, primarily facilitates initial feature extraction within the backbone network of YOLOv8. However, in the context of small target detection, it encounters challenges due to the diverse and intricate background information present in target scenes, as well as the propensity for small targets to be occluded or overlapped by other objects. Consequently, Conv often fails to retain critical information regarding small targets, attributed to its limited generalization capabilities in feature extraction and inadequate local receptive fields. To enhance the model's feature extraction proficiency for small targets in suboptimal environments, this paper introduces the MConv convolution module.

Initially, the input features undergo channel adjustment through convolution operations utilizing a 1×1 kernel size, followed by preliminary feature transformation. These features are then processed through a Batch Normalization (BN) layer and an activation function to yield intermediate features. Subsequently, the intermediate features are subjected to convolution operations with a 3×3 kernel size and processed through another BN layer to expedite network convergence. The resulting features are then residually connected to the original intermediate features. Ultimately, the output features are derived by passing through the activation function once more. The structure of the MConv convolution module is illustrated in Fig 3.

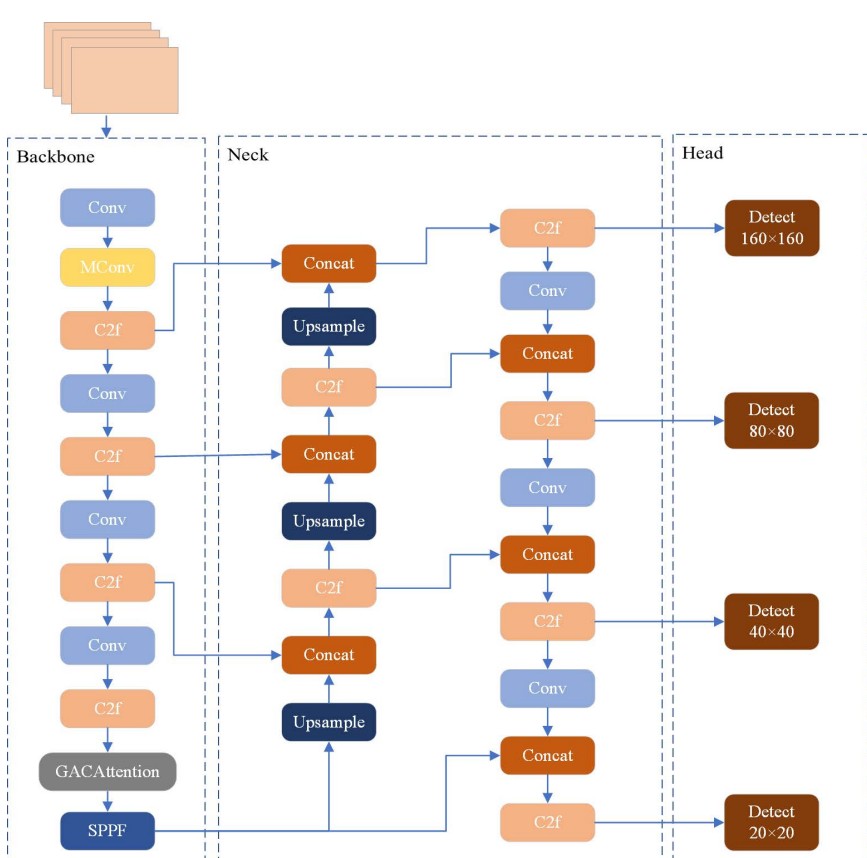

**Fig 2. MGAC-YOLO structure diagram.**

The specific computational process is delineated as follows:

$$F = \text{SiLU}(\text{BN}(\text{Conv}_1(F_{in})))$$

(1)

$$F_{out} = \text{SiLU}(\text{BN}(\text{Conv}_3(F)) + F)$$

(2)

In the equations, $F_{in}$ denotes the input features, SiLU represents the activation function, $\text{Conv}_i$ indicates the convolution operation with a kernel size of i × i, and $F_{out}$ signifies the output features.

The MConv convolution module is designed with a two-layer convolution architecture, which enhances the network's depth, allowing it to capture both detailed and global features across multiple scales, thereby improving the network's feature representation capabilities. Additionally, a multi-level residual connection structure is established, maintaining a consistent computational load throughout the module during training.

## The GACAttention attention module

In computer vision applications, attention mechanisms are typically integrated into models to amplify their focus on critical information within the input data, thereby enhancing model performance and accuracy. For instance, Channel Attention [27] emphasizes the significance of various feature channels to bolster important features while diminishing the influence of less relevant ones, whereas Spatial Attention [27] concentrates on the importance of different spatial locations within an image, enabling the network to dynamically modulate feature responses.

UAV datasets offer diverse scenes and perspectives for object detection tasks. However, their high vulnerability to environmental influences, minor features in the images, significant scale variations, and intricate, dense scenes lead to the gradual loss of small target features during multiple feature extraction and downsampling processes, ultimately impacting object detection accuracy. To address this issue, this paper designs a cross-dimensional attention mechanism named GACAttention, which integrates the global channel attention from GAM [17] and the local spatial attention from CloAttention [18]. This module enables simultaneous focus on global channel relationships and local spatial details, enhancing the model's ability to capture features of small targets. It processes the input feature maps through distinct channel and spatial attention branches to capture more cross-channel sensitive information and derive weight values across various receptive fields. The resulting outputs are then concatenated and subjected to dimensionality reduction and fusion via a convolution module to produce the final output feature maps.

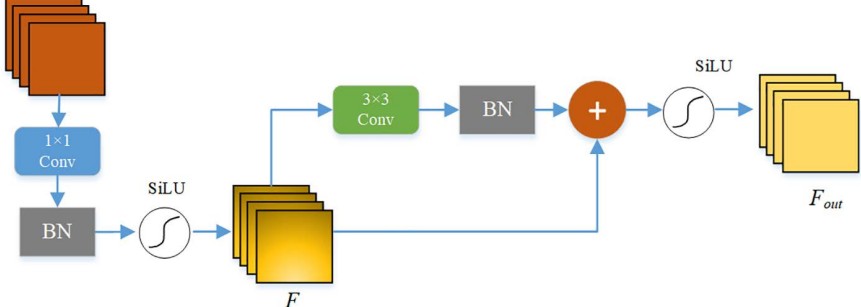

**Fig 3. MConv structure diagram.**

## GAM

The GAM (Global Attention Mechanism) aims to enhance object detection performance by effectively capturing global information. It adaptively adjusts attention across different feature regions, thereby emphasizing significant target features while suppressing irrelevant or distracting information. The overall structure of the GAM is depicted in Fig 4.

From a network architecture standpoint, the GAM mechanism is primarily executed through two essential sub-modules: the channel attention sub-module and the spatial attention sub-module. The channel attention sub-module preserves and enhances the inter-dimensional channel-space dependencies by employing three-dimensional configurations and a two-layer Multi-Layer Perceptron (MLP) [28]. Meanwhile, the spatial attention sub-module integrates spatial information via convolutional layers, deliberately avoiding max-pooling operations to minimize information loss.

From the perspective of the implementation methodology, the GAM employs the "channel first, then space" attention module sorting technique derived from CBAM [29], while also reengineering the sub-modules. In the channel attention module, for the specified input feature $F_{in} \in R^{C \times H \times W}$, global average pooling and max pooling are executed in the spatial dimension to derive two channel representations of $1 \times 1 \times C$. These representations are subsequently processed through a two-layer MLP with shared parameters to generate two features, which are then summed. Following this, the channel weight coefficient $M_c$ is computed using the *Sigmoid* activation function. The element-wise multiplication of $M_c$ and the original feature $F_{in}$ is performed to yield the intermediate state $F$, represented as

$$F = M_c(F_{in}) \otimes F_{in} \tag{3}$$

In the spatial attention module, $F$ undergoes average pooling and max pooling in the channel dimension to produce two channel representations of $H \times W \times 1$. These representations are concatenated along the channel dimension and processed through a $7 \times 7$ convolutional layer for the fusion of spatial information. The spatial weight coefficient $M_s$ is then derived via the *Sigmoid* activation function. The element-wise multiplication of $M_s$ and $F$ is executed to generate the output feature $F_{out}$, indicated as

$$F_{out} = M_S(F) \otimes F \tag{4}$$

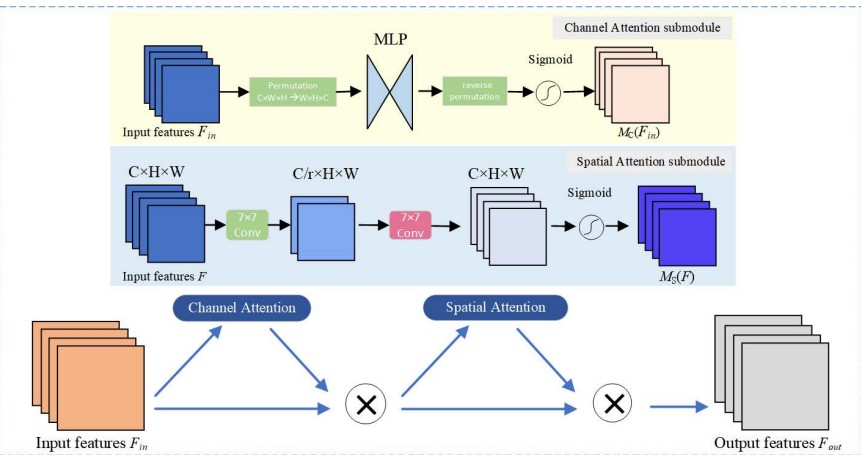

**Fig 4. GAM structure diagram.**

Through the reengineering of both the channel attention and spatial attention sub-modules within GAM, it effectively emphasizes significant features while diminishing less important ones, thereby enhancing performance consistently across various neural network architectures and depths.

## CloAttention

CloAttention (Contextualized Local and Global Attention) is a component of CloFormer [18], developed by a team from Tsinghua University. This attention mechanism integrates the strengths of convolutional operations and attention mechanisms, facilitating the capture of high-frequency local information in images. The network architecture of CloAttention is illustrated in Fig 5.

From a structural perspective, CloAttention employs a "global-local" dual-branch architecture, subsequently fusing the outputs from each branch through a fusion unit. The global branch primarily utilizes standard attention operations to capture low-frequency global information, while the local branch integrates shared weights and context-aware weights to enhance local features, with the objective of extracting high-resolution local feature representations rich in detail. The structural design of CloAttention allows for a more comprehensive handling of image information while maintaining a lightweight profile.

In terms of implementation, the global branch analyzes the entire input data from a macroscopic viewpoint to extract global contextual information. Initially, downsampling is applied to $K$ and $V$, followed by standard attention processing on $Q$, $K$, and $V$ to capture the overall distribution and trends within the input data, thereby obtaining low-frequency global information, as described as follows:

$$X_{global} = \text{Attention}(Q_g, \text{Pool}(K_g), \text{Pool}(V_g))$$

(5)

Conversely, the local branch concentrates on capturing local detail information within the input data. It employs a gating mechanism to generate context-aware weights and introduces enhanced nonlinearity, aiming to better manage the relationships among various positions in the image while preserving the translation equivariance characteristic of convolution. Initially, the same linear transformations utilized in standard attention mechanisms are employed to derive $Q$, $K$, and $V$, as illustrated below:

$$Q, K, V = \text{FC}(X_{in})$$

(6)

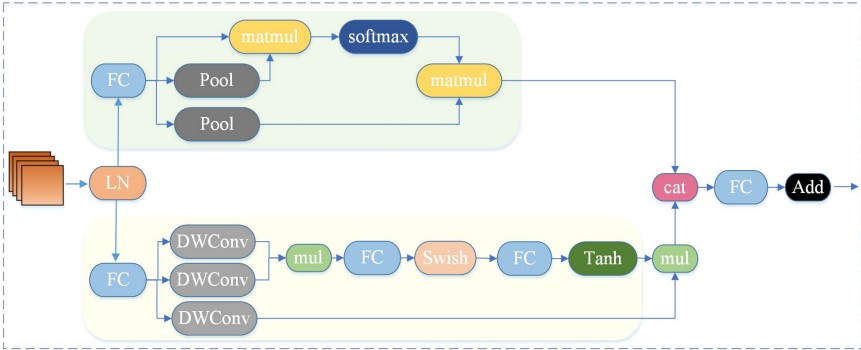

**Fig 5. CloAttention structure diagram.**

In this equation, $X_{in}$ denotes the input features, while FC refers to the fully connected layer. Subsequently, depth-wise convolution (DWConv) [30] with shared weights is applied to $V$ to consolidate local information, represented as follows:

$$V_s = \text{DWConv}(V) \tag{7}$$

DWConv is then utilized to aggregate local information for both $Q$ and $W$, followed by the computation of their Hadamard product. A series of transformations are executed on the resultant output, incorporating Swish and Tanh functions to enhance nonlinearity, thereby yielding higher-quality context-aware weights within the range of −1–1. Ultimately, the generated weight values are employed to augment the local features.

$$Q_l = \text{DWConv}(Q)$$
$$K_l = \text{DWConv}(K) \tag{8}$$

$$Attn_t = \text{FC}(\text{Swish}(\text{FC}(Q_l \odot K_l))) \tag{9}$$

$$Attn = \text{Tanh}(Attn_t / \sqrt{d}) \tag{10}$$

$$X_{local} = Attn \odot V_s \tag{11}$$

In this equation, $d$ signifies the number of channels, and $\odot$ represents the Hadamard product. The output information, processed through both the global and local branches, is concatenated along the channel dimension, culminating in a final output after passing through a fully connected layer.

$$X_t = \text{Concat}(X_{local}, X_{global}) \tag{12}$$

$$X_{out} = \text{FC}(X_t) \tag{13}$$

## Design of the GACAttention module

The GACAttention module innovatively integrates the advantages of the GAM and CloAttention mechanisms to achieve comprehensive feature capture and enhancement. Its structure is illustrated in Fig 6.

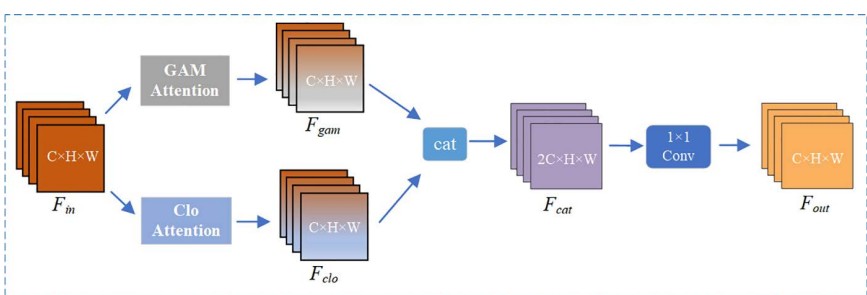

**Fig 6. GACAttention structure diagram.**

From a structural perspective, the GACAttention module operates in three stages: dual-branch independent processing, channel concatenation, and dimensionality reduction with fusion, achieving efficient collaboration between global channel attention and local spatial attention. First, the input features are processed separately through the GAM module and the CloAttention module, generating intermediate features that retain the original channel dimension. The outputs from both branches are then concatenated along the channel dimension, resulting in a combined feature map with dimensions 2C × H × W to preserve complementary information. Finally, a 1 × 1 convolution is applied to compress the channel dimension back to the original input channel count C, followed by grouped convolutions (with groups = C) to enable inter-channel communication while maintaining channel independence, thereby enhancing the flexibility of feature fusion.

$$F_{gam} = \text{GAMAttn}(F_{in}) \tag{14}$$

$$F_{clo} = \text{CloAttn}(F_{in}) \tag{15}$$

$$F_{cat} = \text{Concat}([F_{gam}, F_{clo}], dim = 1) \tag{16}$$

$$F_{out} = \text{Conv}^{group}{}_1(\text{Conv}_1(F_{cat})) \tag{17}$$

In the formula, $F_{in}$ denotes the input features, GAMAttn and CloAttn represent the GAM and CloAttention mechanisms, respectively. $F_{gam}$ and $F_{clo}$ correspond to the intermediate features obtained after processing by the two attention modules. $F_{cat}$ and $F_{out}$ correspond to the features obtained after concatenation and the final output features, respectively. $\text{Conv}^{group}$ refers to grouped convolution, while $\text{Conv}_i$ indicates a convolution operation with an i × i kernel.

By incorporating the GACAttention mechanism into the YOLOv8 benchmark network, the proposed algorithm enhances its ability to focus on and extract features from both local and global perspectives, thereby improving the network's performance in downstream tasks and enhancing the feature extraction capabilities for small targets, ultimately boosting detection performance.

### The Small Target Detection Layer (STDL)

The foundational architecture of YOLOv8 comprises three detection layers, with detection dimensions of 80 × 80, 40 × 40, and 20 × 20, respectively. Typically, small target detection datasets for unmanned aerial vehicles feature relatively diminutive small targets, with dense and complex scene elements, resulting in limited available information. Furthermore, the downsampling factors in YOLOv8 are considerably large, making it challenging for deeper feature maps to capture the feature information of small targets, which can lead to missed detections and adversely affect detection efficacy. To address this issue, this paper introduces a small target detection layer with a dimension of 160 × 160, based on the original algorithm, to capture the feature information of shallower small targets. This enhancement allows for the fusion of shallower feature maps with deeper feature maps for improved detection performance of small targets. The feature extraction structure following the addition of the small target detection layer is depicted in Fig 7.

## Experiments and results analysis

### Dataset and experimental environment

The experiment utilizes the publicly available VisDrone2019 [31] dataset, which is extensive in scale and encompasses urban and rural environmental scenes across 14 different cities in China from the perspective of UAVs. It includes 10 predefined categories: pedestrians, people, cars, vans, buses, trucks, motorcycles, bicycles, awning tricycles, and tricycles. This dataset contains a large number of images captured by UAVs. Due to factors such as shooting distance and angles,

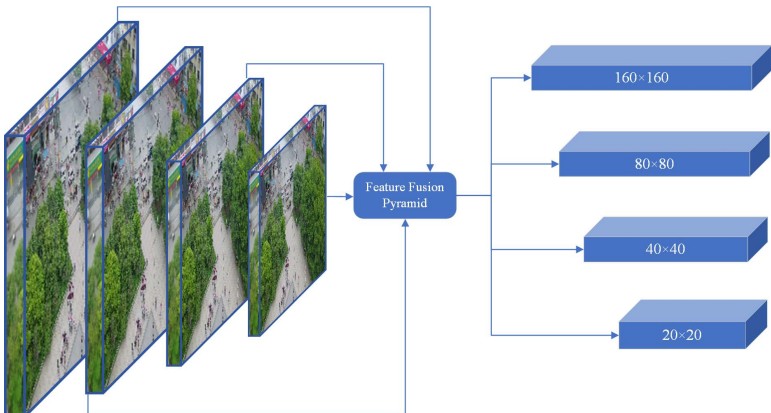

**Fig 7. Structure diagram of the improved feature extraction.**

these images include numerous small-sized targets (e.g., pedestrians, bicycles) that occupy a small number of pixels and exhibit low-resolution characteristics in the images. The VisDrone2019 dataset comprises 6,471 images in the training set, 548 images in the validation set, and 1,610 images in the test set. The scenes within the dataset are realistic, featuring a relatively high number of small targets and an imbalanced distribution among categories, thus fulfilling the requirements for training and evaluating small target detection models.

The Manjaro Linux operating system has been chosen as the experimental platform, featuring an NVIDIA GeForce RTX™ 4090 graphics card. The environment is configured with Python 3.12.5, PyTorch 2.4.1, and CUDA 12.3, with specific training parameter settings summarized in Table 1. Model training employs Stochastic Gradient Descent (SGD) for optimization. Convergence is observed after 300 training epochs, with an initial learning rate of 0.01 and a momentum of 0.937, to ensure training stability.

## Evaluation metrics

To accurately assess the model's performance, this experiment employs precision ($P$), recall ($R$), and mean average precision ($mAP$) metrics, utilizing Intersection over Union (IoU) thresholds of 50% and an average across the range of 50% to 90%. Precision measures the ratio of correctly identified targets by the model to the total number of targets, emphasizing the accuracy of target identification. Recall, on the other hand, quantifies the ratio of correctly identified targets to the total number of actual targets, focusing on the model's identification capability. The specific calculation formulas are as follows:

$$P = \frac{TP}{FP + TP}$$

(18)

**Table 1. Training parameter configuration.**

| Parameter | Value |
|---|---|
| Batch size | 16 |
| Epochs | 300 |
| Image size (px) | 640 |
| Optimizer | SGD |
| Initial learning rate (lr0) | 0.01 |
| Momentum | 0.937 |
| Weight_decay | 0.0005 |

$$R = \frac{TP}{TP + FN} \tag{19}$$

Here, $TP$ denotes the number of true positives identified by the model, $FP$ indicates the number of false positives, and $FN$ represents the false negatives. Average Precision ($AP$) refers to the mean precision for a specific category within the dataset at varying recall levels, while $mAP$ assesses the average precision performance across all categories. The calculation formula is as follows:

$$mAP = \frac{1}{n} \sum_{i=1}^{n} AP_i \tag{20}$$

## Ablation Experiments and Analysis

To illustrate the beneficial effects of the MConv convolutional module, the GACAttention attention mechanism, and the small target detection layer proposed in this study on network enhancement, ablation experiments are conducted to validate the efficacy of each module. The experiments utilize the VisDrone2019 dataset, employing the YOLOv8s network architecture as the baseline model, sequentially integrating the GACAttention module, the MConv module, and the small target detection layer. Each of these combined models undergoes training for 300 epochs, with results presented in Table 2.

In the data presented in the table, Model 1 corresponds to the baseline YOLOv8s model. Model 2 reflects the model after incorporating the dual attention fusion module GACAttention into the backbone network. Compared to Model 1, it demonstrates increases in the Precision, Recall, $mAP_{50}$, and $mAP_{50-95}$ by 1.3%, 0.3%, and 0.2%, respectively, indicating that the GACAttention module effectively extracts features of small targets from both global and local perspectives, thereby enhancing the model's feature processing capabilities for small targets. Model 3 represents the model that substitutes Conv with the MConv designed in this study, building upon Model 1. In comparison to Model 1, the Precision, Recall, $mAP_{50}$, and $mAP_{50-95}$ have improved by 1.7%, 0.9%, 1%, and 0.6% respectively. This enhancement suggests that the multi-layer convolutional architecture and residual connections incorporated in the MConv convolutional module proposed in this study allow the model to extract more comprehensive feature information across various dimensions, effectively addressing the issues of vanishing and exploding gradients, thereby enhancing detection capabilities. Model 4 further refines the model by simultaneously employing both the GACAttention and MConv modules, resulting in superior experimental outcomes compared to Models 2 and 3. When juxtaposed with the baseline model, the Precision, Recall, $mAP_{50}$, and $mAP_{50-95}$ have increased by 2.1%, 1.1%, 1.1%, and 0.9% respectively. The experimental findings indicate that integrating feature information from multiple channels and dimensions enables the model to acquire richer feature representations while simultaneously bolstering its robustness. Model 5 introduces a small target detection layer on top of Model 4, which further enhances the Precision, Recall, $mAP_{50}$, and $mAP_{50-95}$ by 3.2%, 4.8%, 5.2%, and 3.5% respectively.

**Table 2. Results of ablation experiments.**

| Model | GACAttention | MConv | STDL | Precision/% | Recall/% | $mAP_{50}$/% | $mAP_{50-95}$/% |
|---|---|---|---|---|---|---|---|
| 1 | – | – | – | 50.2 | 38.5 | 39.5 | 23.7 |
| 2 | √ | – | – | 51.5 | 38.4 | 39.8 | 23.9 |
| 3 | – | √ | – | 51.9 | 39.4 | 40.5 | 24.3 |
| 4 | √ | √ | – | 52.3 | 39.6 | 40.6 | 24.6 |
| 5 | √ | √ | √ | 55.5 | 44.4 | 45.8 | 28.1 |

The addition of the small target detection layer allows the improved algorithm presented in this paper to capture feature information of small targets within a shallower network. Given that the features of small targets in shallower networks are relatively abundant, merging this feature information with the semantic data from deeper networks effectively mitigates the issue of missed detections. Ablation studies confirm that the MConv and GACAttention attention modules, along with the newly introduced small target detection layer, are all beneficial to varying degrees.

To illustrate the model's improvement more clearly, a visual comparison of key evaluation metrics between the enhanced model MGAC-YOLO and YOLOv8s has been conducted. The results of this comparison are depicted in Fig 8. The data presented in the figure indicates that, under identical iteration counts, the experimental results for MGAC-YOLO in metrics such as $mAP_{50}$, $mAP_{50-95}$, and Recall surpass those of the baseline model, demonstrating the proposed algorithm's notable advantages.

## Comparative experiments and analysis

To validate the performance of the MGAC-YOLO algorithm introduced in this paper for small target detection, a comparative analysis with other leading algorithms was performed under consistent experimental conditions, parameter settings, and utilizing the VisDrone2019 dataset. The results of these comparative experiments are summarized in Table 3.

The data presented in the table clearly indicates that our algorithm has achieved notable advancements in small target detection when compared to the Faster-RCNN and SDD model algorithms. Specifically, the $mAP_{50}$ and $mAP_{50-95}$ metrics have improved by 9.7%, 24.4%, 6.5%, and 14.8%, respectively. In comparison to the YOLO series, including YOLOv5s

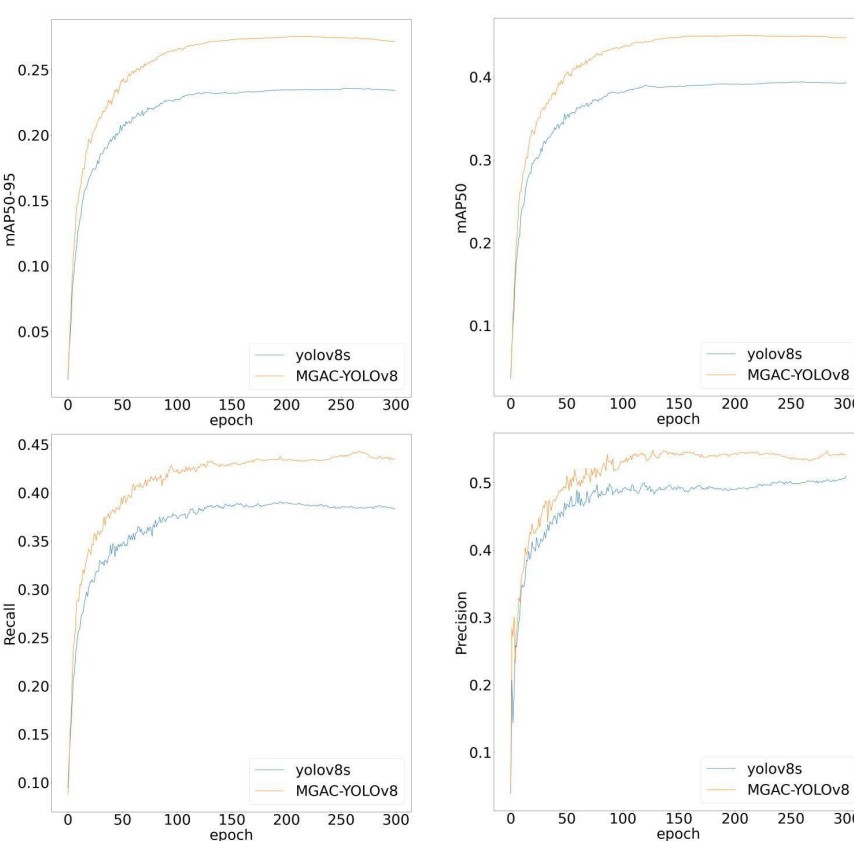

**Fig 8. Visual comparison chart of training results.**

**Table 3. Results of comparative experiments.**

| model | mAP$_{50}$/% | mAP$_{50-95}$/% | Precision/% | Recall/% | Number of parameters |
|---|---|---|---|---|---|
| Faster-RCNN | 36.1 | 21.6 | – | – | 42.36 |
| SDD | 21.4 | 13.3 | – | – | 26.13 |
| YOLOv5s | 38.9 | 23.4 | 50.5 | 37.7 | 9.12 |
| YOLOv8s | 39.5 | 23.7 | 50.2 | 38.5 | 11.13 |
| Gold-YOLO [32] | 40.5 | 23.8 | 50.6 | 38.6 | 21.5 |
| LW-YOLO [33] | 42.9 | 26.3 | 52.1 | 41.3 | 5.58 |
| YOLOv9c | 43.9 | 26.8 | 55.8 | 42.3 | 25.33 |
| MGAC-YOLO | 45.8 | 28.1 | 55.5 | 44.4 | 12.36 |

and YOLOv8s, our algorithm demonstrates superior detection capabilities for small targets, with increases of 6.3% and 6.9% in mAP$_{50}$, and enhancements of 4.7% and 4.4% in mAP$_{50-95}$. When evaluated against the YOLOv9c model algorithm, our algorithm maintains a significant advantage, with mAP$_{50}$ and mAP$_{50-95}$ values exceeding those of YOLOv9c by 1.7% and 1.3%, respectively, while also possessing a considerably lower parameter count. In relation to the Gold-YOLO algorithm, our algorithm has achieved increases of 5.3% and 4.3% in mAP$_{50}$ and mAP$_{50-95}$, respectively, with the parameter count of Gold-YOLO being approximately double that of our algorithm. Although our improved algorithm has a slightly lower parameter count than the LW-YOLO (Lightweight YOLO) algorithm, it has nonetheless achieved increases of 2.9% and 1.8% in mAP$_{50}$ and mAP$_{50-95}$, respectively, compared to LW-YOLO. Overall, the MGAC-YOLO algorithm proposed in this study exhibits distinct advantages in small target detection relative to other leading algorithms.

## Visual comparison and analysis

To provide a more intuitive representation of the small target detection capabilities of the improved MGAC-YOLO algorithm from the perspective of unmanned aerial vehicles, we selected images from various scenes within the VisDrone2019 dataset to evaluate the model's detection performance under diverse conditions, including variations in lighting, occlusion, and angles, as illustrated in Figs 9–12. In nighttime scenarios, despite uneven lighting and instances where certain objects are obscured, overlapped, or significantly occluded, the model presented in this study is still able to accurately detect a variety of target types. In low-light environments characterized by shadows and dynamic blurring, the proposed model effectively identifies target categories. In densely populated scenes, where target overlap and occlusion are prevalent, leading to the potential loss of target features, the model is capable of detecting and accurately classifying the majority of targets. For high-altitude images commonly encountered in unmanned aerial vehicle target detection, the number of

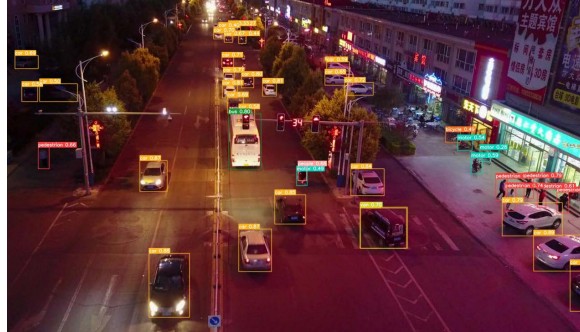

**Fig 9. Detection result of MGAC-YOLO in dark scenes.**

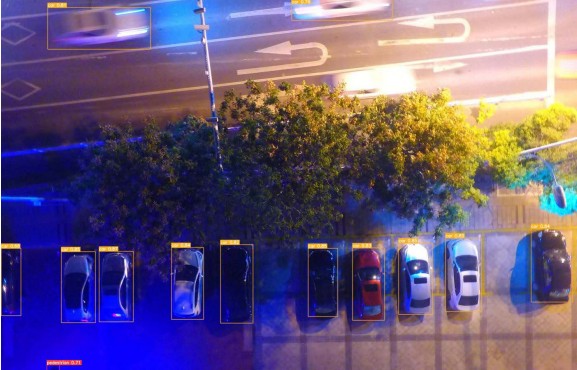

**Fig 10. Detection result of M-YOGACLOv8 under conditions of shadows and dynamic blur.**

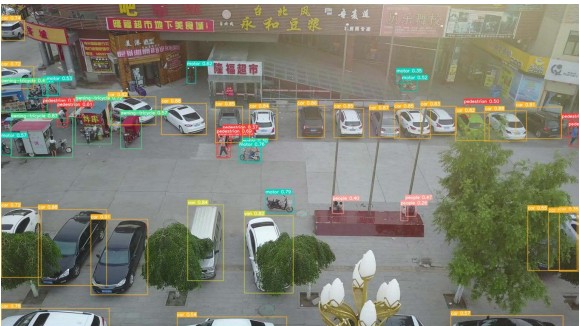

**Fig 11. Detection result of MGAC-YOLOv8 in occluded scenes.**

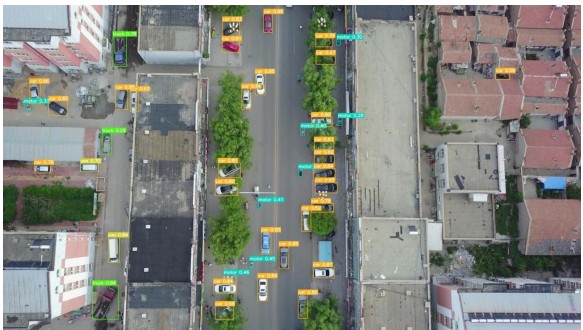

**Fig 12. Detection result of MGAC-YOLOv8 in high-altitude photography.**

visible targets is significantly reduced, and the abundance of background information, influenced by shooting angles and distances, often results in a scarcity of target feature information. Nevertheless, the detection outcomes indicate that the improved algorithm presented in this paper has substantially enhanced detection performance for small targets.

To clearly demonstrate the algorithmic improvements, we conducted comparative experiments on the baseline model YOLOv8s and the proposed MGAC-YOLO using identical image sets, as illustrated in Fig 13. As can be seen from images a1 and b1, in the scenes with long distances and occlusions, the improved model MGAC-YOLO, with the help of

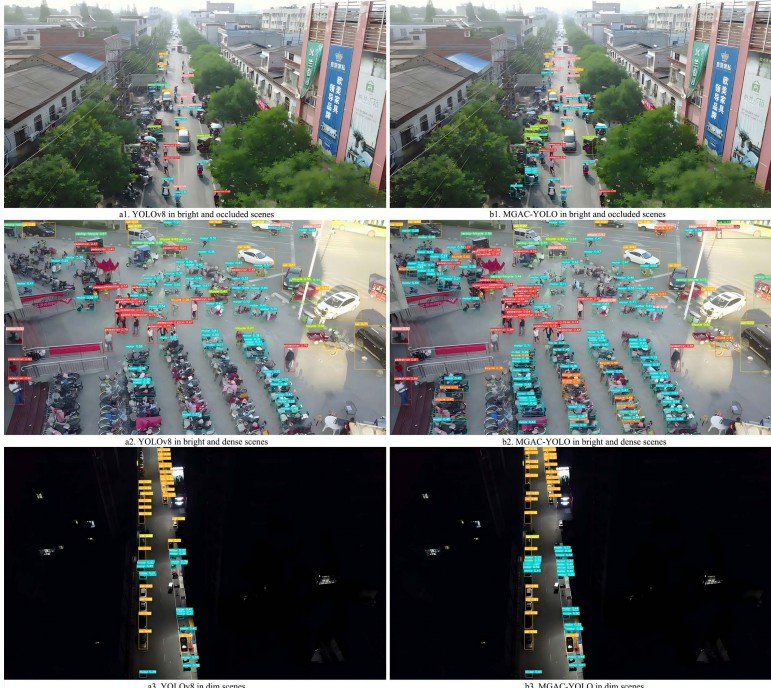

**Fig 13. Visual comparative experiment.**

the MConv structure and the GACAttention module, enhances the ability to capture the features of small targets at long distances. At the same time, it focuses on the key features of the targets to reduce background interference, and both the detection rate and accuracy for various targets are higher than those of the baseline model. From images a2 and b2, it can be seen that in the scenes where the target types are similar and overlapping, the collaboration between the GACAttention module and the small target detection layer enables precise discrimination of subtle feature differences, achieving a notable improvement in detection precision.. As can be seen from images a3 and b3, under low-light conditions, MGAC-YOLO exhibits superior ability to extract and utilize faint features, detecting significantly more targets than the baseline.These results show that MGAC-YOLO outperforms YOLOv8s in detecting small targets across diverse complex scenarios.

## Conclusion

This paper addresses the challenges of omission and misdetection in small target detection by unmanned aerial vehicles by proposing an enhanced algorithm, MGAC-YOLO, with YOLOv8s serving as the baseline model. Initially, the standard convolution (Conv) is redesigned by increasing the convolutional depth and incorporating residual connections. This modification strengthens feature extraction capabilities while maintaining a controlled increase in computational complexity. Furthermore, by integrating the global attention mechanism of GAM and the local feature capture advantage of CloAttention, the proposed GACAttention structure achieves cross-dimensional information interaction, enabling richer feature representation for small targets. Lastly, a small target detection layer is introduced, which fuses shallow-layer detail features with deep-layer semantic features, thereby enhancing the model's detection performance for small targets.. Experimental results on the VisDrone2019 dataset demonstrate that the improved algorithm achieves higher detection accuracy and improved mAP values compared to the baseline model and other mainstream models, validating the effectiveness of the

proposed method. Notably, despite the proposed MConv structure and GACAttention module effectively limiting parameter growth (total parameters increased by approximately 11% compared to YOLOv8s) through residual connections and light-weight design, the introduction of multi-layer convolutions and new modules results in a slight increase in inference time versus the original model. For practical deployment, hardware acceleration techniques (e.g., TensorRT) are recommended to further optimize runtime efficiency. Future work will focus on investigating lightweight feature fusion strategies and exploring model compression and quantization techniques to balance performance and real-time requirements. Additionally, subsequent plans aim to extend the improved approach to feature fusion networks, integrating more efficient model architecture designs to further enhance the practicality and robustness of UAV-based small target detection.

## Author contributions

**Conceptualization:** Xiujing Li, Haifei Zhang, Yiliu Hang, Hao Chen.

**Data curation:** Xiujing Li, Haifei Zhang, Yiliu Hang, Hao Chen.

**Formal analysis:** Xiujing Li, Haifei Zhang, Yiliu Hang, Hao Chen.

**Investigation:** Xiujing Li, Haifei Zhang, Yiliu Hang, Hao Chen.

**Methodology:** Xiujing Li, Haifei Zhang, Yiliu Hang, Hao Chen.

**Project administration:** Xiujing Li, Haifei Zhang, Yiliu Hang, Hao Chen.

**Resources:** Xiujing Li, Haifei Zhang, Yiliu Hang, Hao Chen.

**Software:** Xiujing Li, Haifei Zhang, Yiliu Hang, Hao Chen.

**Validation:** Xiujing Li, Haifei Zhang, Yiliu Hang, Hao Chen.

**Visualization:** Xiujing Li, Haifei Zhang, Yiliu Hang, Hao Chen.

**Writing – original draft:** Xiujing Li, Haifei Zhang, Yiliu Hang, Hao Chen.

**Writing – review & editing:** Xiujing Li, Haifei Zhang, Yiliu Hang, Hao Chen.

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
