## [Decision Letter · Decision Letter 0]

Dear Dr. LI,

Thank you for submitting your manuscript to PLOS ONE. After careful consideration, we feel that it has merit but does not fully meet PLOS ONE’s publication criteria as it currently stands. Therefore, we invite you to submit a revised version of the manuscript that addresses the points raised during the review process.

We look forward to receiving your revised manuscript.

Kind regards,

Chenchu Xu, Ph.D

Academic Editor

PLOS ONE

Journal Requirements:

3. In the online submission form, you indicated that the data that support the findings of this stuty are avalible from the corresponding author upon resonable request. 

4. Please remove your figures from within your manuscript file, leaving only the individual TIFF/EPS image files, uploaded separately. These will be automatically included in the reviewers’ PDF.

Additional Editor Comments:

The paper on MGAC-YOLO for UAV small target detection shows promise but requires revision. The innovation is unclear, and the authors need to deepen their analysis of current technologies, clarify the definition of small targets, improve image quality, and provide stronger experimental evidence to support claims of enhanced accuracy and balanced inference speed.

Reviewers' comments:

Reviewer's Responses to Questions

**Comments to the Author**

1. Is the manuscript technically sound, and do the data support the conclusions?

Reviewer #1: Yes

Reviewer #2: Yes

2. Has the statistical analysis been performed appropriately and rigorously?

Reviewer #1: No

Reviewer #2: Yes

3. Have the authors made all data underlying the findings in their manuscript fully available?

Reviewer #1: Yes

Reviewer #2: Yes

4. Is the manuscript presented in an intelligible fashion and written in standard English?

Reviewer #1: No

Reviewer #2: Yes

Reviewer #1: This paper addresses the challenges of omission and misdetection in small target detection by unmannedaerial vehicles by proposing an enhanced algorithm, MGAC-YOLO, with YOLOv8s serving as the baseline model.

However, the innovation of this paper is not clear.

Some comments are given to improve the paper quality.

A. This paper lacks a deep understanding of the difficulties in existing technologies for drone target recognition. In fact, the problem discussed in this paper already has a large number of feasible results.

B. The image quality needs improvement.

C. The definition of small target objects in this paper is confusing, and a strict definition is needed for this issue.

D. The experimental results reveal that, in comparison to other leading algorithms, the improved algorithm exhibits enhanced detection accuracy and mAP values, while also achieving a balance between detection accuracy and inference speed. The results presented in this paper do not effectively support this conclusion.

Reviewer #2: The manuscript presents a novel algorithm, MGAC-YOLO, for small target detection in UAV applications, demonstrating improvements over existing models. The article has a well-structured format, and the language and article structure are generally good. However, there are some minor problems to be addressed, below are detailed recommendations for improvement.

**Do you want your identity to be public for this peer review?** For information about this choice, including consent withdrawal, please see our Privacy Policy

Reviewer #1: No

Reviewer #2: **Yes: ** I agree to disclose my name as the reviewer in the published peer review history.

---

## [Author Response · Author response to Decision Letter 1]

7 May 2025

Dear Editor and Reviewers,

Thank you for your thorough review and constructive feedback on our manuscript titled "Small Target Detection Algorithm Based on the Fusion Attention Mechanism and Multi-Layer Convolution". We have carefully addressed all feedback and revised the manuscript accordingly. The modifications are highlighted in the revised manuscript using tracked changes or annotations for clarity. We kindly request your further review of the updated version. Below are our point-by-point responses to each comment。

Response to Reviewer #1

Comment A: "This paper lacks a deep understanding of the difficulties in existing technologies for drone target recognition. In fact, the problem discussed in this paper already has a large number of feasible results."

Response: We have reviewed recent literature closely related to this study and updated the Introduction section (Paragraph 2, page 3 of the revised manuscript) with a concise discussion of existing research achievements, while explicitly outlining remaining technical challenges. Additionally, the innovations of MGAC-YOLO have been rearticulated in the revised manuscript (page 4). All specific modifications are highlighted using tracked changes and annotations for clarity.

Comment B: "The image quality needs improvement."

Response: We have reprocessed the images used in the manuscript. In the revised manuscript, Figures 9 and 10 have been updated by replacing the original low-resolution images with high-quality versions to ensure clarity in visual presentation.

Comment C: "The definition of small target objects in this paper is confusing, and a strict definition is needed for this issue."

Response: We have added a detailed description of the definition of small targets in the Introduction section (page 4, paragraph 1 of the revised manuscript). Furthermore, in the "Dataset and Experimental Environment" section (page 12), we explicitly state that the selected VisDrone2019 dataset aligns with the small target criteria defined in this study. These revisions are marked with tracked changes for clarity.

Comment D: "The experimental results reveal that, in comparison to other leading algorithms, the improved algorithm exhibits enhanced detection accuracy and mAP values, while also achieving a balance between detection accuracy and inference speed. The results presented in this paper do not effectively support this conclusion."

Response: This study prioritizes improving detection accuracy as a core research focus, with the effectiveness of each proposed module validated through ablation experiments and comparative experiments (detailed in the "Experiments and Results Analysis" section). Regarding the original statement on "balancing detection accuracy and inference speed", we identified a misrepresentation in the initial conclusion, as no quantitative comparison or optimization of inference speed was conducted. Consequently, the revised manuscript removes these unsubstantiated claims, retaining only experimentally supported results on accuracy improvement. Additionally, the Conclusion section now includes a discussion on the real-time performance limitations of the improved model, enhancing the practical applicability of the research. Future work will focus on exploring lightweight attention mechanisms and model compression techniques (e.g., knowledge distillation, model quantization) to reduce computational complexity while maintaining detection accuracy.

Response to Reviewer #2

Comment 1: "The description of the GACAttention module's integration strategy is not clearly, and experimental hyperparameters (e.g., learning rate decay strategy) is inadequate. More technical details are needed to improve the study's reproducibility."

Response: We have revised the "GACAttention Module" section of the manuscript to provide detailed enhancements to the integration strategy of the GACAttention module. In the updated content (pages 7–12 of the revised manuscript), we first introduce the design motivation for the GACAttention module, then explain the implementation principles of GAM and CloAttention individually, followed by a description of the GACAttention module structure and integration strategy, and finally discuss the positive impacts of integrating GACAttention on model performance. Regarding the experimental hyperparameters, we have refined their descriptions in the "Dataset and Experimental Environment" section and added "Table 1 Training Parameter Configuration" (page 13 of the revised manuscript), which explicitly lists the values of all hyperparameters to ensure reproducibility and transparency.

Comment 2: It is recommended to add a discussion on the model's computational efficiency or real - time limitations in the conclusion section. This will enhance the study's completeness and practicality.

Response: In the Conclusion section, we have expanded the discussion on the real-time performance limitations of the model in practical applications. Specifically, we note that the introduction of multi-layer convolutions and new modules has led to a approximately 11% increase in total parameters (compared to YOLOv8s) and higher inference latency. To address these challenges, we propose potential optimization directions, such as lightweight attention mechanisms and model compression techniques. Despite these real-time constraints, the model demonstrates significant improvements in small target detection accuracy. We believe that future enhancements in computational efficiency and other aspects will further expand its applicability in related domains.

Comment 3: "Definitions and formatting in formulas (1) and (2) are non-standard. eg: "SiLu" and "SiLU" are used inconsistently."

Response: We have standardized the use of "SiLU" throughout the manuscript, revised the symbols in Equations (1) and (2) (page 7 of the revised manuscript), and meticulously checked and corrected definition errors and formatting inconsistencies in other equations. All modifications are highlighted using tracked changes in the revised manuscript for clarity.

Comment 4: "The figures are numbered in confusion, such as: the subfigures of Figure 10 are not clearly referenced in the text."

Response: We have systematically verified all figure and table citations in the manuscript to ensure accurate references within the main text. Specifically, we revised the visual comparative analysis section by adding in-text citations and detailed descriptions for each subfigure of Figure 10 (a-f). Following these revisions, the team conducted cross-checking to guarantee that all figure references are accurate and consistent. The finalized updates are marked with tracked changes in the revised manuscript.

Comment 5: "Some references are missing volume numbers or page numbers (e.g., Reference [7], [11]). Please conduct a thorough check of the reference format."

Response:We have revised all references in accordance with PLOS ONE's formatting guidelines, supplementing missing information and correcting formatting inconsistencies. These updates are marked in the revised manuscript for review.

Comment 6: "The manuscript inconsistently uses "YOLOv8" and "YOLOV8", requiring standardization to "YOLOv8" for professional consistency."

Response: We have checked and corrected the spelling errors throughout the text, and uniformly used "YOLOv8".

Response to academic editor

Comments: The paper on MGAC-YOLO for UAV small target detection shows promise but requires revision. The innovation is unclear, and the authors need to deepen their analysis of current technologies, clarify the definition of small targets, improve image quality, and provide stronger experimental evidence to support claims of enhanced accuracy and balanced inference speed.

Response:Thank you for recognizing the research potential of our manuscript and providing valuable revision suggestions. We have comprehensively addressed every comment and revised the manuscript accordingly. The specific modifications are outlined below:

1. Enhanced Analysis of Existing Technologies and Innovation Highlights

We reviewed recent literature closely related to this study and updated the "Overview" section (page 3 of the revised manuscript) with a brief discussion of existing research achievements. We also listed remaining technical challenges, such as insufficient cross-scale interaction leading to inadequate extraction of small target details, feature confusion in densely occluded scenarios, and information loss during downsampling. Based on this, the innovation points of this paper have been re-condensed as follows: an improved residual convolution module that enhances the feature expression ability by increasing the depth and residual connections (without significantly increasing the computational load); designing the GACAttention mechanism by combining the advantages of GAM and CloAttention to achieve cross-channel feature interaction; and adding a dedicated detection layer for small objects to improve the recall rate of small objects through shallow feature fusion.

2.Clarification of Small Target Definition

We added a definition of small targets in the "Overview" section (page 4, paragraph 1 of the revised manuscript): "Small targets refer to objects that occupy a small number of pixels in the image or are small in size relative to the entire image and exhibit low-resolution characteristics."

We also confirmed in the "Dataset and Experimental Environment" section (page 12 of the revised manuscript) that the VisDrone2019 dataset aligns with this definition.

3. Improved Image Quality

We reprocessed all figures in the manuscript. Figures 9 and 10 now use high-resolution images to replace the original low-resolution versions.

4.Strengthening of Experimental Evidence

This paper focuses on the improvement of detection accuracy as the research direction, and the effectiveness of each improved module has been verified through ablation experiments and comparative experiments (see Table 3 and Figure 7 for details). Regarding the expression of "balancing detection accuracy and inference speed", there was a deviation in the original conclusion. In fact, there was no quantitative comparison and optimization of the inference speed, so the relevant misrepresentation has been deleted. At the same time, a discussion on the real-time limitations has been supplemented, and a plan for further optimization by combining technologies such as TensorRT during actual deployment has been proposed, and the subsequent research direction of "exploring lightweight attention mechanisms and model compression techniques" has been clarified.

All revisions are marked with tracked changes in the "Revised Manuscript with Track Changes." We believe the revised version meets the publication requirements of PLOS ONE and respectfully request your further review.

Thank you again for your time and expert advice!

Sincerely,

Xiujing LI

---

## [Decision Letter · Decision Letter 1]

Small target detection algorithm based on the fusion attention mechanism and multi-layer convolution

PONE-D-24-59247R1

Dear Dr. LI,

We’re pleased to inform you that your manuscript has been judged scientifically suitable for publication and will be formally accepted for publication once it meets all outstanding technical requirements.

Kind regards,

Chenchu Xu, Ph.D

Academic Editor

PLOS ONE

Additional Editor Comments (optional):

Reviewers' comments:

Reviewer's Responses to Questions

**Comments to the Author**

Reviewer #1: All comments have been addressed

Reviewer #2: All comments have been addressed

2. Is the manuscript technically sound, and do the data support the conclusions?

Reviewer #1: Yes

Reviewer #2: Yes

3. Has the statistical analysis been performed appropriately and rigorously?

Reviewer #1: Yes

Reviewer #2: Yes

4. Have the authors made all data underlying the findings in their manuscript fully available?

Reviewer #1: Yes

Reviewer #2: Yes

5. Is the manuscript presented in an intelligible fashion and written in standard English?

Reviewer #1: Yes

Reviewer #2: Yes

Reviewer #1: (No Response)

Reviewer #2: (No Response)

**Do you want your identity to be public for this peer review?** For information about this choice, including consent withdrawal, please see our Privacy Policy

Reviewer #1: No

Reviewer #2: No

---

## [Editor Report · Acceptance letter]

PONE-D-24-59247R1

PLOS ONE

Dear Dr. LI,

I'm pleased to inform you that your manuscript has been deemed suitable for publication in PLOS ONE. Congratulations! Your manuscript is now being handed over to our production team.

Kind regards,

on behalf of

Dr. Chenchu Xu

Academic Editor

PLOS ONE